# Detection of *Giardia duodenalis* Zoonotic Assemblages AI and BIV in Pet Prairie Dogs (*Cynomys ludovicanus*) in Bangkok, Thailand

**DOI:** 10.3390/ani12151949

**Published:** 2022-07-31

**Authors:** Ketsarin Kamyingkird, Pornkamol Phoosangwalthong, Nutsuda Klinkaew, Alisara Leelanupat, Chanya Kengradomkij, Wissanuwat Chimnoi, Teerapat Rungnirundorn, Burin Nimsuphan, Tawin Inpankaew

**Affiliations:** 1Department of Parasitology, Faculty of Veterinary Medicine, Kasetsart University, Lad Yao, Chatuchak, Bangkok 10900, Thailand; ketsarinkamy@hotmail.com (K.K.); eve2155082@gmail.com (P.P.); fvetnuk@ku.ac.th (N.K.); alis.maymee@gmail.com (A.L.); fvetcyk@ku.ac.th (C.K.); wiss_chim@hotmail.com (W.C.); 2Exotic Clinic, Kasetsart University Veterinary Teaching Hospital, Chatuchak, Bangkok 10900, Thailand; estel1819@hotmail.com

**Keywords:** giardiasis, genetic characterization, pet prairie dogs, zoonosis

## Abstract

**Simple Summary:**

Prairie dogs are native to the United States, Canada, and Mexico. They have been brought to Thailand and become popular as exotic pets and providing humans with exotic pets in close-contact environments. Prairie dogs have been known to carry several waterborne protozoan pathogens. One of those pathogens is known as *Giardia*, a flagellate protozoan parasite which can cause gastrointestinal illness in humans and other animals. This study has identified the *G**iardia* parasite in the feces of prairie dogs. There were 13% of prairie dogs pets in Bangkok, Thailand that carried the *Giardia* parasite. We also found that the parasite was categorized as a human parasite. Therefore, there was potential risk that prairie dogs could contract a *Giardia* infection from humans as a source, and that humans could receive the parasite from their exotic pets via the fecal-oral route in shared environments. We suggest that exotic pet owners should pay more attention to effective sanitation and provide clean food and water for their exotic pets. Owners should bring their exotic pets for veterinary services and screening of zoonotic pathogens using fecal examination regularly. Treatment can successfully cure the infected pets as well as prevent the spreading of pathogen to the environment.

**Abstract:**

*Giardia* is a flagellate protozoa that can be transmitted via direct contact and by consuming contaminated water. It is pathogenic in humans and various other animals, including exotic pets. Pet prairie dogs are popular in Thailand, but they have not been investigated regarding giardiasis. *Giardia* infection was measured, and genetic characterization was performed to investigate the zoonotic potential of *Giardia* carried by pet prairie dogs. In total, 79 fecal samples were examined from prairie dogs visiting the Kasetsart University Veterinary Teaching Hospital during 2017–2021. Simple floatation was conducted. Two *Giardia*-positive samples were submitted for DNA extraction, PCR targeting the *Giardia*
*ssu rRNA*, *tpi* and *gdh* genes was performed, and genetic characterization using sequencing analysis was conducted. Risk factors associated with *Giardia* infection were analyzed. *Giardia* infection was found in 11 out of the 79 pet prairie dogs (13.9%). *Giardia* infection was significantly higher in male prairie dogs (*p* = 0.0345). Coccidia cysts (12.7%), the eggs of nematodes (6.3%), and amoeba cysts (2.5%) were also detected. Genetic characterization of the two *Giardia*-positive samples revealed that they were *G. duodenalis* assemblage A, sub-genotypes AI and assemblage B, and sub-genotype BIV, the zoonotic assemblages. This was the first report of *Giardia* infection in pet prairie dogs in Bangkok, Thailand. The results revealed that these pet prairie dogs in Thailand were infected with zoonotic assemblages of *G. duodenalis* sub-genotype AI, which might have been derived from animal contaminants, whereas sub-genotype BIV might have been derived from human contaminants. Owners of prairie dogs might be at risk of giardiasis or be the source of infection to their exotic pets.

## 1. Introduction

Herbivorous-burrowing rodents in the genus *Cynomys*, known as prairie dogs, are native to the United States, Canada, and Mexico [1]. Five species of prairie dogs have been documented: *Cynomys ludovicanus*, the black-tailed prairie dog; *C. pardivens*, the Utah prairies dog; *C. mexicanus*, the Mexican prairie dog; *C. leucurus*, the white-tailed prairie dog; and *C. gunnisoni*, Gunnison’s prairie dog [2]. In nature, intestinal parasitic diseases in prairie dogs are common [3]. The black-tailed prairie dog is the most common species kept in captivity in zoos as well as in private homes as an exotic pet [1]. In the current decade, prairie dogs have become popular as exotic pets in Thailand. Prairie dogs eat plants and insects and belong to the squirrel family (Sciuridae) of rodents (order Rodentia); they are at risk of contracting protozoan pathogens via the consumption of contaminated food and water. This may lead to gastrointestinal parasites and protozoan infection in prairie dogs. 

Giardiasis is caused by an intestinal flagellate enteric protozoan parasite in the genus *Giardia* that infects humans and animals. The *Giardia* life cycle involves two stages, the first as a trophozoite and then as a cyst [4]. The *Giardia* cyst is responsible for disease transmission through the ingestion of contaminated water and food [5,6]. *Giardia* caused weight loss, loose stools, and diarrhea in a laboratory colony of prairie dogs [7]. *Giardia duodenalis* (synonym *G. intestinalis* and *G. lamblia*) [8], *G. muris,* and *G. microti* have been reported in animals related to prairie dogs, such as rodents [9]. *Giardia duodenalis* is a broad host-range parasite that can infect domestic and wild mammals, including humans [10,11,12,13]. Therefore, the infection of *G. duodenalis* in prairie dogs may indicate that they are a possible wildlife reservoir or the victims of pathogen spill-over [7]. *Giardia* parasites can be identified by their trophozoites and/or cysts via microscopic examination from fecal or intestinal contents. Polymerase chain reactions (PCRs) were also developed for more sensitive and specific identification of *Giardia* species. Genetical differentiation using several genetic markers, including *β-giardin* (*bg*), *triose phosphate isomerase* (*tpi*), *the small subunit ribosomal RNA* (*ssu rRNA*), and *glutamate dehydrogenase* (*gdh*), has helped to identify at least eight assemblages (A–H) of *G. duodenalis*, with *G. duodenalis* assemblage G identified from rodents [9,14].

In Thailand, giardiasis has been reported in humans [15,16,17,18] and in many animals, including dogs [19,20], cats [21] and long-tailed macaques [22]. However, information on the prevalence of *Giardia* infection in captive prairie dogs in Thailand is still limited. Therefore, the objective of the current study was to determine *Giardia* infection in prairie dogs in Thailand and to genetically characterize pet prairie dogs infected with *Giardia* to reveal the risk of zoonotic parasites among exotic pets and their owners.

## 2. Materials and Methods

### 2.1. Origin of Samples and Screening of Giardia

This study was conducted based on 79 fecal samples collected from pet prairie dogs visiting the Exotic Clinic, Kasetsart University Veterinary Teaching Hospital during 2017–2021. Fecal samples were collected during the clinical examination then submitted to the Department of Parasitology for screening of *Giardia* and other intestinal parasites using simple floatation. *Giardia*-positive samples were selected for molecular diagnosis.

### 2.2. Identification of Giardia Using Molecular Diagnosis

#### 2.2.1. Genetic Characterization of *Giardia*


*Giardia*-positive fecal samples were tested for DNA extraction using a stool DNA extraction kit (Flavorgen, Pingtung, Taiwan), then stored at −20 °C until use. Identification of *Giardia* was conducted using nested PCR methods. Three sets of primers were used—the *small subunit ribosomal RNA* gene (*ssu rRNA*), the *triose phosphate isomerase* gene (*tpi*), and the *glutamate dehydrogenase* gene (*gdh*), as shown in Table 1.

#### 2.2.2. Detection of *Giardia ssu rRNA* Gene

The amplification of the *ssu rRNA* gene was performed using a nested PCR as previously described [23,24]. Briefly, PCR reactions were prepared in total volumes of 25 µL, consisting of 10× PCR buffer 2.5 µL, 25 mM MgSO_4_ 1.5 µL, 10 mM dNTPs 0.5 µL, 10 pmol of primers 1 µL, 0.5 unit of Taq DNA polymerase 0.5 µL (ABM Good, Richmond, Canada), 5% dimethyl sulfoxide (DMSO) 1 µL (DMSO, Carl Roth, Karlsruhe, Germany), and 2 µL of template genomic DNA. Sterile distilled water was used as a negative control and a positive control was extracted from a *G. duodenalis* trophozoite-positive fecal sample. A Mastercycler nexus PCR thermal cycler (Eppendorf, Hamburg, Germany) was used for PCR amplification. The first PCR amplification consisted of initial degeneration at 94 °C for 3 min, followed by 35 cycles of degeneration at 94 °C for 45 s, annealing at 58 °C for 45 s and extension at 72 °C for 1 min, with a final extension for 1 cycle at 72 °C for 3 min. The nested PCR conditions were same as the first conditions.

#### 2.2.3. Detection of *G. duodenalis tpi* Gene

The amplification of the *tpi* gene (530 bp) was conducted using a nested PCR as previously described [25]. The nested PCR was prepared in a total volume of 25 µL as mentioned above. The first PCR amplification consisted of initial degeneration at 96 °C for 5 min, followed by 35 cycles of degeneration at 96 °C for 45 s, annealing at 50 for 30 s and extension at 72 °C for 45 s, with a final extension for 1 cycle at 72 °C for 7 min. The nested PCR conditions were as mentioned above except that the annealing temperature was 55 °C.

#### 2.2.4. Detection of *G. duodenalis gdh* Gene

The amplification of the *gdh* gene (530 bp) was performed using a nested PCR as previously described [26] in a total volume of 25 µL PCR reaction, as mentioned earlier. The primary and secondary amplification consisted of initial degeneration at 94 °C for 2 min, followed by 35 cycles of degeneration at 94 °C for 30 s, annealing at 50 °C for 30 s and extension at 72 °C for 1 min, with a final extension for 1 cycle at 72 °C for 7 min [26]. Gel electrophoresis was performed to visualize the PCR product size from each PCR product, using 1.5% agarose gel and 100 V for 40 min and UV transluminescence. 

**Table 1 animals-12-01949-t001:** List of primers used for genetic characterization of *Giardia* in prairie dog fecal samples.

Primer Name (Sequences)	Target Gene	Product Size	Reference
TPI-F3: 5′-AAATIATGCCTGCTCGTCG-3′	*tpi*	835 bp	[25]
TPI-R3: 5′-CAAACCTTITCCGCAAACC-3			
TPI-F4: 5′-CCCTTCATCGGIGGTAACTT-3′		530 bp	
TPI-R2: 5′-GTGGCCACCACICCCGTGCC-3′			
RH1 1 5′-CATCCGGTCGATCCTGCC-3′	*ssu rRNA*	140 bp	[23]
RH4 5′-AGTCGAACCCTGATTCTCCGCCAGG-3′			
Giar-F 5′-GACGCTCTCCCCAAGGAC-3′		292 bp	[24]
Giar-R 5′-CTGCGTCACGCTGCTCG-3′			
Gdh1 5′-TTCCGTRTYCAGTACAACTC-3′	*gdh*	754 bp 530 bp	[26]
Gdh2 5′-ACCTCGTTCTGRGTGGCGCA-3′			
Gdh3 5′-ATGACYGAGCTYCAGAGGCACGT-3′			
Gdh4 5′-GTGGCGCARGGCATGATGCA-3′			

### 2.3. DNA Sequencing and Bioinformatic Analysis

The PCR products of the *ssu rRNA*, *tpi*, and *gdh* genes were purified using a gel extraction kit (Flavorgen, Pingtung, Taiwan). The purified PCR products were submitted to MACROGEN, South Korea for DNA sequencing. The purity of the DNA sequences was qualified by visualization using Unipro UGENE software (V.41.0.), as previously documented [27], and compared with the GenBank database in the National Center for Biotechnology Institute (NCBI) using Basic Local Alignment Search Tools (BLAST).

### 2.4. Statistical Analysis

Sex and age from clinical record were evaluated in terms of its association with *Giardia* positivity in pet prairie dogs using Fisher’s exact test. Statistical significance was tested at *p* > 0.05. 

## 3. Results

### 3.1. Occurrence of Giardia and Other Intestinal Parasites in Pet Prairie Dogs

*Giardia* infection in pet prairie dogs was 13.9% (11/*79*) based on simple floatation. *Giardia* infection was significantly dominant (*p* = 0.0345) in males (22.9%, 8/35) compared to females (6.8%, 3/44) (Table 2). Among the positive samples of *Giardia* spp., only the cyst stage was found with an average size of approximately 8.7 × 12 µm, with an oval shape and a smooth, thin wall. Cysts filled with a larva of protozoa that consisted of an axostyle and four nuclei were visualized and indicated as immature *Giardia* cysts based on microscopic examination (Figure 1). The number of immature *Giardia* cysts varied among the positive samples (from low to adequate cyst numbers). Two positive fecal samples based on microscopic examination with adequate numbers of *Giardia* cysts (approximately 3–5 cysts/viewed at 10× magnification) were selected for molecular diagnosis and genetic characterization. 

In addition, the eggs of nematodes and cysts of amoeba were detected in the fecal samples of the pet prairie dogs (Table 2). The intestinal parasites infection was 31.6% (25/79). Intestinal parasites were found in male and female prairie dogs (31.4% (11/35) and 31.8% (14/44), respectively). The intestinal parasite infections were 35.9% (14/39) in young and 20.0% (8/40) in adult prairie dogs. There were 12.7% (10/79) coccidia, 6.3% (4/79) nematodes, and 6.3% (4/79) amoeba infections (Table 2). Two prairie dogs had mixed infections of *Giardia* with nematodes and coccidia with nematodes. Interestingly, coccidia infection in the pet prairie dogs was dominant among young prairie dogs (23.1%, 9/39) compared to adults (2.5%, 1/40).

### 3.2. Molecular and Genetic Characterization of Giardia in Pet Prairie Dogs

The nested PCR and sequencing analysis of the *ssu rRNA* gene confirmed that the *Giardia* found in two pet prairie dogs were *G. duodenalis* with 99.21–100% identity. Amplification of the *tpi* gene using a nested PCR confirmed that one prairie dog was infected with *G. duodenalis* assemblage A, sub-genotype AI (100% similarity) and another prairie dog was infected with *G. duodenalis* assemblage B, sub-genotype BIV (98.77% similarity). Sequencing of the *gdh* gene in each sample also confirmed the same assemblages with 99.20–100% similarity. All the *ssu rRNA*, *tpi* and *gdh* partial sequences derived from the two samples in this study were submitted to GenBank with accession numbers ON025978, ON025979, ON037509, ON037510, ON037511, and ON037512, respectively (Table 3). The bioinformatic analysis results indicated that *G. duodenalis* sub-genotype AI might have derived from animal contaminants, whereas sub-genotype BIV might have derived from human contaminants.

## 4. Discussion

Giardiasis is distributed worldwide. The surveillance of giardiasis in the USA between 2009 and 2010 revealed almost 20,000 cases annually, especially in children aged 1–9 years [28]. Laboratory identification of *Giardia* can be primarily detected using microscopic examination, rapid diagnostic assays, fluorescent microscopy, and molecular-based assays, with high sensitivity and specificity [5]. In addition, giardiasis in humans has been associated with major outbreaks in domestic animals, household pets, and farm animals [5]. Wild prairie dogs are common in urban landscapes of the USA and Canada; however, they have been introduced by humans into tropical countries, contributing to increasing concern of the potential dispersal of pathogens from exotic pets to humans and other livestock animals [29]. Prairie dogs have been known to carry several water-borne protozoan pathogens [30], including *Giardia*. In recent times, prairie dogs have been brought to Thailand as exotic pets. 

The prevalence of *Giardia* in humans was reported to vary in the range 0–73.4% in the Southeast Asia, and the Far East [16]. In Thailand, the prevalence levels of *G. duodenalis* infection in humans varies in different provinces: 1.25% in Ayudthaya and Suphanburi [31], 2.21% in Chiang Mai [32], 2.2% in Surin, 6.5% in Samut Sakhon [33], 13.6–23.3% in Kanchanaburi [34], 17.6% in Bangkok [35], and 37.7% in Pathum Thani [15]. Similarly, the prevalence levels of *G. duodenalis* in animals in Thailand varies: 3.0–25.2% in dogs [20,36], 27.3% in cats [36], 1.6% in water buffaloes [37], and 7.0% in long-tailed macaques [22]. However, epidemiological study of *Giardia* infection in captive prairie dogs is very rare. The current results revealed that *Giardia* infection was predominant in pet prairie dogs followed by coccidia, nematode parasites, and amoeba, respectively. *Giardia* infection in pet prairie dogs in the current study was similar to the studies of [34,35] in humans and in dogs [36]. However, to compare the prevalence of *Giardia* in pet prairie dogs with such other studies is difficult due to the limited research conducted.

Pet prairie dogs have adapted from the wild, where they live in colonies under sociable conditions. Being kept as a solitary house-pets may lead to high stress and increased susceptibility to diseases, especially in young animals that have been separated from their mother, or in male prairie dogs that normally need to be surrounded by several females in the colony [38]. In the current study, the prevalence of *Giardia* was significantly higher in young, male pet prairie dogs, which might relate to the reasons described above. The rate was also similar to those in other reports that indicated significantly higher levels of *Giardia* infection in young, poor, and immunodeficient humans, especially in urban areas [16]. 

The current study used microscopic examination, which is the gold standard technique, in combination with molecular assays to achieve high sensitivity and specificity, as recommended [6]. A nested PCR targeting the *tpi* and *gdh* gene targets was useful and helped to confirm issues in the the pet prairie dogs in the current study, such as infection with zoonotic species of *G. duodenalis*, assemblage A and B. *G. duodenalis* assemblages A and B have been most frequently reported in humans as well as in cattle, dogs, and cats from different parts of the world [5]. In addition, *G. duodenalis* assemblages A and B have been found in prairie dogs captured from field sites [7,38]. *Giardia duodenalis* assemblage A consists of two distinct clusters, AI and AII, whereas assemblage B consists of sub-genotypes BIII and BIV [39]. In Thailand, *G. duodenalis* sub-genotypes AI, AII, BIII, and BIV have been reported [35]. However, AII was the most dominant sub-genotype in Thailand, [35]. The current study identified the sub-genotypes AI and BIV in pet prairie dogs, similar to other reports. In addition, *G. duodenalis* infection in humans causes gastrointestinal clinical manifestations, including diarrhea, vomiting, abdominal cramps, and general malaise [5]. Clinical symptoms of diarrhea were recorded in some prairie dogs with *Giardia* infection [7]; however, most of the prairie dogs in our study had no specific clinical signs. 

*Giardia duodenalis* assemblage A and B have been identified in non-human primates [22,40], which are closely related to humans. This is possibly the source of *Giardia* infection for other animals that share the same habitat and environment. Prairie dogs could be potential wildlife reservoirs, or the victims of pathogen spill-over as has been reported [7]. However, *G. muris* and *G. microti* seem to be the predominant species of *Giardia* in wild rodents, not *G. duodenalis* [9]. Some studies in China and Russia identified intestinal parasites in house-kept pets, and house-kept and zoo animals, based on genotyping analysis that indicated there was the possibility of pets, and house-kept and zoo animals receiving human-derived *G. duodenalis* isolates [30,41]. Thus, it might be that prairie dogs might contract *G. duodenalis* infection through the consumption of contaminated water and food supplied by humans. Therefore, food and water sanitization must be practiced by the pet owners to prevent giardiasis. A limitation of the current study was the small prairie dog population sample used to perform molecular identification. This was due to the low number of fecal samples received from the Kasetsart University Veterinary Teaching Hospital. To better understand the diversity of *Giardia* species infection in prairie dogs, a larger prairie dog population should be examined based on the surveillance of food and water-borne zoonotic protozoa in pet prairie dogs or from exotic animal farms. The origin of *G. duodenalis* infection in prairie dogs in Thailand is still not clear regarding whether the pathogen is transmitted from human sources or comes from animal sources. More *Giardia* isolates from prairie dogs should be used for further studies of genetic characterization. In addition, future study should involve the evaluation of food and water management and the detection of *Giardia* in both human owners and their exotic pets, which would help to identify the source of *G. duodenalis* transmission between humans and exotic pets.

## 5. Conclusions

This study was the first retrospective study on *Giardia* infections in prairie dog pets in Bangkok, Thailand. Genetic characterization of two *Giardia*-positive samples revealed that they were *G. duodenalis* assemblage A, sub-genotypes AI and assemblage B and sub-genotype BIV, the zoonotic assemblages. Prairie dogs might have become infected with *G. duodenalis* through the consumption of contaminated water and food supplied via human resources. Therefore, improved food and water sanitization must be a priority for the owners of pet prairie dogs to prevent giardiasis.

## Figures and Tables

**Figure 1 animals-12-01949-f001:**
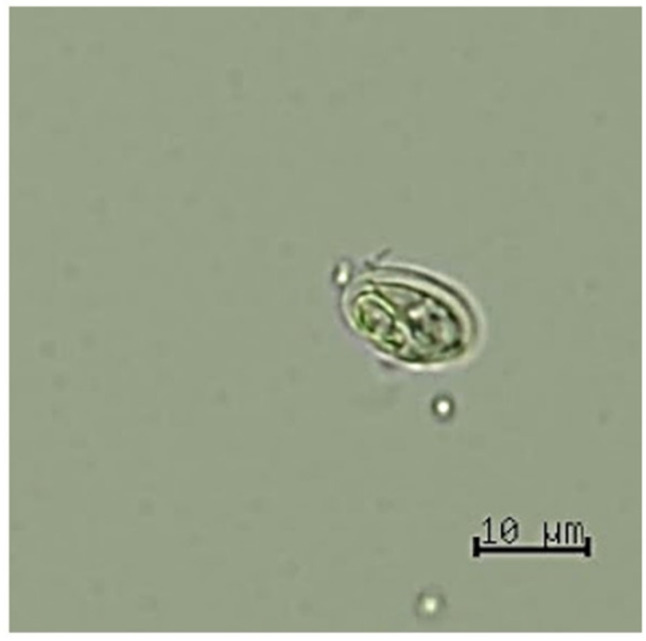
Cyst of *Giardia duodenalis* in prairie dog fecal sample.

**Table 2 animals-12-01949-t002:** *Giardia* and other intestinal parasites infections in pet prairie dogs based on microscopic examination.

Factor		Total Number	Positive Number				
			Total Intestinal Parasites (%)	*Giardia* spp. (%)	Coccidia (%)	Nematodes (%)	Amoeba (%)
Gender	Female	44	14 (31.8%)	3 (6.8%)	5 (11.4%)	1 (2.3%)	1 (2.3%)
	Male	35	11 (31.4%)	8 (22.9%) *	5 (14.3%)	3 (8.6%)	1 (2.9%)
Age	<6 months	39	14 (35.9%)	5 (12.8%)	9 (23.1%)	4 (10.3%)	1 (2.6%)
	>6 months	40	8 (20.0%)	6 (20.0%)	1 (2.5%)	0 (0.0%)	1 (2.6%)
Total		79	25 (31.6%)	11 (13.9%)	10 (12.7%)	4 (5.1%)	2 (2.5%)

* *p*-value < 0.05.

**Table 3 animals-12-01949-t003:** Genetic characterization of *G. duodenalis* in pet prairie dogs.

Gene Targets	Prairie Dog 1 (Isolate: PS61-623)		
	Identified Species (Assemblage, Sub-Genotype)	Similarity (%)	Submitted to GenBank with Accession No.
*ssu rRNA*	*G. duodenalis* (NA)	100%	ON025978
*tpi*	*G. duodenalis* (assemblage B, BIV)	98.77%	ON037509
*gdh*	*G. duodenalis* (assemblage B, BIV)	99.20%	ON037511
**Gene Targets**	**Prairie Dog 2 (Isolate: PS61-629)**		
*ssu rRNA*	*G. duodenalis* (NA)	99.21%	ON025979
*tpi*	*G. duodenalis* (assemblage A, AI)	100%	ON037510
*gdh*	*G. duodenalis* (assemblage A, AI)	100%	ON037512

NA = Not Available.

## Data Availability

Not applicable.

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
