# Peer review of "Detection of Giardia duodenalis Zoonotic Assemblages AI and BIV in Pet Prairie Dogs (Cynomys ludovicanus) in Bangkok, Thailand"

_animals, 2022, doi:10.3390/ani12151949_

Round 1
Reviewer 1 Report
1. Is it perhaps possible to enlarge the study sample?
2. Are there giardia infecìtions in captive rodent pets?
3. Although it is a poorly known topic, some strains of Giardia can infect both humans and animals, including various pets, as chinchillas (Chincilla laniger), beavers (Castor sp.), birds (sp. plurimae), opossums (sp. plurimae), and monkeys (sp. plurimae). The topic could be quite relevant for the management of the sanitary system pets/children, particularly in Western Europe, where Giardiasis at present is not widespread in humans.
4. It is a remarkable contribution to a scarcely considered topic. The only doubt concerns the possibility that Giardia infection in the prairie dogs could be due to local diffusion of Giardiasis rather than to the commerce of prairie dogs pets. It is a relevant question because while Giardiasis is widespread in Thailand, it is not in Europe, where however prairie dogs are quite diffuse as pets Its role as reservoir and diffusor of Giardiasis in Thailand and in Europe is surely dissimilar
Author Response
Dear respected reviewer
We appreciate the reviewer's comments and suggestions.
We have noted as the reviewer's commented here;
- Is it perhaps possible to enlarge the study sample?
Answer: We apologize for not being able to enlarge the study sample as the reviewer’s commented. It is due to the fact that we have not recruited fecal samples from prairie dogs, but we have received the samples from the exotic unit for diagnostic purposes. Moreover, only a small volume of feces was received and not enough for DNA extraction/ PCR diagnosis.
- Are there giardia infections in captive rodent pets?
Answer: There is no investigation of Giardia infections in captive rodent pets in Thailand yet.
- Although it is a poorly known topic, some strains of Giardia can infect both humans and animals, including various pets, as chinchillas (Chincilla laniger), beavers (Castor sp.), birds (sp. plurimae), opossums (sp. plurimae), and monkeys (sp. plurimae). The topic could be quite relevant for the management of the sanitary system pets/children, particularly in Western Europe, where Giardiasis at present is not widespread in humans.
Answer: Thank you for the reviewer comments. We understand that the reviewer suggested changing the title of the study to relevant management and sanitary systems to prevent Giardia widespread. In this study we have not collected data relevant to water management and the sanitary procedures from prairie dogs owners, only detection of the pathogen from prairie dogs was performed. It is the first evidence of zoonotic Giardia infection in pet prairie dogs in our country which may lead to the future study on management and the sanitary system. We have already mentioned ‘In addition, future study should involve the evaluation of food and water management and the detection of Giardia in both human owners and their exotic pets would help to identify the source of G. duodenalis transmission between humans and exotic pets.’ in the discussion part.
- It is a remarkable contribution to a scarcely considered topic. The only doubt concerns the possibility that Giardia infection in the prairie dogs could be due to local diffusion of Giardiasis rather than to the commerce of prairie dogs pets. It is a relevant question because while Giardiasis is widespread in Thailand, it is not in Europe, where however prairie dogs are quite diffuse as pets Its role as reservoir and diffusor of Giardiasis in Thailand and in Europe is surely dissimilar.
Answer: Thank you for the reviewer’s comments. We have changed the topic to ‘The zoonotic assemblages AI and BIV of Giardia duodenalis, were detected in pet prairie dogs (Cynomys ludovicanus) in Bangkok, Thailand’. as the reviewers commented. We agree that Giardia infection in the prairie dogs could be due to local diffusion of Giardiasis rather than to the commerce of prairie dogs pets. Since prairie dogs were imported to Thailand as pets about 10 years ago.

Reviewer 2 Report
Authors aims to determine the prevalence of Giardia infection in praire dogs in Thailand and to genetically characterize pet prairie dogs infected with Giardia, revealing the risk of zoonotic parasites among exotic pets and their owners.
Specific comments/suggestions are highlighted in the attached .pdf file.
More details must be included in the M&M section regarding sample size calculation and the sampling process. Why only two samples were genetically analyzed? And why the statistical analysis using Fisher's exact test, it is necessary to present more information about what factors were analyzed and how were they collected.
Table 2 must be corrected to inform clearly what each line means, if each row is the comparison between sex, and why the p-values are reported in sex vs coccidia if the manuscript is about Giardia.
It is important to clarify these point, even when the topic is becoming more and more important each day, due to the increased report of Giardia in pets, dogs and cats, and the increased resistance to potential treatments. Also an increase in exotic or non-traditional susceptible pet species could play an important role in the transmission and maintenance of this zoonotic parasite.

Author Response
Dear Respective Reviewer
Thank you very much for your comments and suggestions.
We have noted to each comments here;
Authors aims to determine the prevalence of Giardia infection in praire dogs in Thailand and to genetically characterize pet prairie dogs infected with Giardia, revealing the risk of zoonotic parasites among exotic pets and their owners.
Answer: Thank you for valuable suggestions. Due to the samples used in this report were collected from prairie dogs visited exotic unit, KU veterinary teaching hospital due to its illness and/ or regular health check. To measure the prevalence might not be suitable for this study. We have revised accordingly.
Specific comments/suggestions are highlighted in the attached .pdf file.
- More details must be included in the M&M section regarding sample size calculation and the sampling process. Why only two samples were genetically analyzed? And why the statistical analysis using Fisher's exact test, it is necessary to present more information about what factors were analyzed and how were they collected.
Answer: Thank you for the reviewer’s valuable comments. Regarding to sample size calculation, we would like to report Giardia identification from the fecal samples that the department of parasitology have received from the exotic unit during 2017-2021 rather than to perform the epidemiological study. Since, there is no information on the number of pets prairie dogs registered in Thailand yet. Hence, there were approximately 57 prairie dog cases visited veterinary teaching hospital annually based on Kasetsart University Veterinary Teaching Hospital records. Not all cases have requested for fecal examination. Therefore, limited number of prairie dog fecal samples were recieved and the sample size was not included.
We have revised the sentence to ‘The fecal samples were collected by picking from the deficated prairie dogs during the clinical examination was performed then submitted to the Department of Parasitology for screening of Giardia and other intestinal parasites using simple floatation.’ in the M&M.
In this study, there were only two samples used for genetic characteization due to small amount of feces were received. The amount of fecal sample was enough for the simple floatation (as the requested method by the veterinarian) but there were not enough for DNA extraction/ PCR diagnosis. That is why only two samples were submitted to molecular diagnosis.
We have used Fisher’s exact test instead of multivariate test and Chi Square test due to the fact that we have less than 50 positive samples in this study, and fisher’s exact test is more reliable for small population.
We have added ‘including sex and age’ in the statistical analysis. Page 4
- Table 2 must be corrected to inform clearly what each line means, if each row is the comparison between sex, and why the p-values are reported in sex vs coccidia if the manuscript is about Giardia.
Answer: We apologize for the error. Thank you for the reviewer's comments.
We have corrected table 2.
We have included the results of coccidia as it was one of the most prevalent pathogens followed by Giardia that is found in prairie dogs. However, we have removed the p-values against coccidia as the reviewer’s commented.
- It is important to clarify these point, even when the topic is becoming more and more important each day, due to the increased report of Giardia in pets, dogs and cats, and the increased resistance to potential treatments. Also an increase in exotic or non-traditional susceptible pet species could play an important role in the transmission and maintenance of this zoonotic parasite.
Answer: Thank you for the valuable comments. We have considered changing the title of our study to ‘The zoonotic assemblages AI and BIV of Giardia duodenalis, were detected in pet prairie dogs (Cynomys ludovicanus) in Bangkok, Thailand’ as the reviewers have remarked.
We have also revised other parts recommended by the reviewer and higlighted in yellow.

Round 2
Reviewer 2 Report
Authors aims to determine the prevalence of Giardia infection in praire dogs in Thailand and to genetically characterize pet prairie dogs infected with Giardia, revealing the risk of zoonotic parasites among exotic pets and their owners.
The authors considered most of the comments and suggestions made by the reviewers. Specific comments/suggestions are highlighted in the attached .pdf file.

Author Response
Dear Respective Reviewer,
We appreciated for our kind suggestion that help to improve this manuscript.
1. We have change the title to 'Detection of Giardia duodenalis zoonotic assemblages AI and BIV in pet prairie dogs (Cynomys ludovicanus) in Bangkok, Thailand' as the reviewer suggested.
2. We have revised the fecal collection to 'Fecal samples were collected from defecated prairie dogs during the clinical examination then submitted to the Department of Parasitology for screening of Giardia and other intestinal parasites using simple floatation.' in 2.1.
3. We have corrected a sentence to 'Sex and age from clinical record were evaluated in terms of its association with Giardia positivity in pet prairie dogs using Fisher's exact test. ' as the reviewer suggested.
4. We have corrected 'used' to 'use' in the institutional review board statement.
All revised highighted in yellow.
We hope for your kind condideration.